**Field experiments in ocean alkalinity enhancement research**
Tyler Cyronak[1]*, Rebecca Albright[2]*, Lennart T. Bach[3]
[1]Georgia Southern University, Institute for Coastal Plain Science, Statesboro, GA
[2]California Academy of Sciences, San Francisco, CA
[3]Institute for Marine and Antarctic Studies, University of Tasmania, Hobart, Tasmania, Australia
*These authors contributed equally to the writing of this manuscript.
*Correspondence to:* Tyler Cyronak (tcyronak@georgiasouthern.edu) and Rebecca Albright
(ralbright@calacademy.org)

## Abstract

This chapter focuses on considerations for conducting open-system field experiments in the context of ocean alkalinity enhancement (OAE) research. By conducting experiments in real-world marine and coastal systems, researchers can gain valuable insights into ecological dynamics, biogeochemical cycles, and the safety, efficacy, and scalability of OAE techniques under natural conditions. However, logistical constraints and complex natural dynamics pose challenges. To date, only a limited number of OAE field studies have been conducted, and guidelines for such experiments are still evolving. Due to the fast pace of carbon dioxide removal (CDR) research and development, we advocate for openly sharing data, knowledge, and lessons learned as quickly and efficiently as possible within the broader OAE community and beyond. Considering the potential ecological and societal consequences of field experiments, active engagement with the public and other stakeholders is desirable while collaboration, data sharing, and transdisciplinary scientific teams can maximize the return on investment. The outcomes of early field experiments are likely to shape the future of OAE research, implementation, and public acceptance, emphasizing the need for transparent and open scientific practices.

## 1. Introduction

This chapter addresses considerations for conducting open-system field experiments related to ocean alkalinity enhancement (OAE). We define 'field experiment' or 'field studies' broadly as the addition or manipulation of alkalinity in a natural system that is relevant to OAE, independent of the spatial and temporal scale. We intentionally exclude spatial and temporal scales from our definition to encompass the wide spectrum of OAE methods and approaches. In fact, field experiments are likely to span spatial scales of $m^2$ to 100s of $km^2$ and last from days to years. Field experiments and studies differ from both 'field trials' and 'field deployments' in their motivation, as both trials and deployments denote the practical application and usage of a specific product, device, or technology. The scientific focus during field trials is likely to be on the efficacy of Carbon Dioxide Removal (CDR) and fine-tuning operational deployment, while field experiments will encompass a broader range of scientific goals and objectives. The nature, logistics, and objectives of field experiments are likely to make them smaller in scale than operational deployments. This will be advantageous, as field experiments that emulate planned OAE trials and deployments will help create the scientific framework needed to scale operational OAE safely and responsibly.

The benefits of conducting experiments in natural systems include observing complex ecological dynamics and impacts at the ecosystem level, understanding the role of biogeochemical cycles and physical processes that cannot be replicated in other settings, and assessing CDR under real-world scenarios. The complexity and breadth of some field experiments will necessitate science that transcends disciplinary boundaries, making collaboration a priority. Success in the field faces many challenges due to the inherent complexity of natural systems along with limiting logistical constraints (e.g., permitting, access, social license, infrastructure, life cycle emissions). Despite these challenges, the first OAE field experiments are already underway, many of which are small-scale representations of scalable OAE approaches. There will be much to learn from these early studies, and any knowledge or insights gained should be shared as efficiently and openly as possible within the wider OAE community and beyond.

While some OAE field experiments have been completed or are already in progress, many more

are on the horizon. We recommend that three overarching questions be taken into consideration,
especially when in the planning stages:
*What are the main goals of the experiment?*
Establishing the objectives of a field experiment early in the planning stage will help guide all
aspects of the scientific research plan, including site selection, measurement techniques and
approaches, data analysis, and measured outcomes. Potential overarching goals of OAE field
experiments include demonstrating functionality, efficacy, process, and/or scalability,
determining ecological and environmental impacts, developing measurement, reporting, and
verification (MRV) protocols, and assessing community engagement. Life cycle assessments
(LCA) may be a critical learning objective for some projects (e.g., Foteinis et al. 2023),
especially those that are examining OAE at the scale of operational deployments. This list of
overarching goals is not comprehensive, and goals are not necessarily mutually exclusive. For
example, larger projects may wish to assess multiple components of an OAE approach while
smaller projects might be highly focused.

*What is the type of alkalinity perturbation?*
The type of alkalinity that is added (e.g., aqueous vs. solid, carbonates, hydroxides, oxides, or
naturally occurring (ultra-)mafic rocks) will ultimately determine many aspects of the scientific
research plan. For example, projects adding ground alkaline minerals (e.g., olivine) to the ocean
may have different goals and timelines than projects that add aqueous alkalinity (e.g., liquid
NaOH) (see Eisaman et al., 2023, this Guide). Priorities for projects adding ground material
might include tracking the dissolution of the alkaline material plus monitoring the fate of the
dissolved alkalinity and its dissolution co-products (e.g. trace metals), while projects adding
aqueous alkalinity will likely be more concerned with the latter. Other important experimental
considerations that will be driven by the type of alkalinity perturbation include the concentration
of added alkalinity, duration of additions, dilution and advection at the field site, residence time,
air-sea equilibration, co-deployed tracers, sampling scheme, and environmental side-effects.
These and other research considerations are discussed in more detail below.
*What are the permitting constraints and wider social implications?*
Addressing the appropriate regulatory requirements is essential before any field experiment can
move forward. Permitting requirements will be influenced by the study location, type of
alkalinity perturbation, spatial scale, and duration. The use of existing infrastructure (e.g.,
wastewater discharge sites) and environmental projects (e.g., beach renourishment) may offer
ways to facilitate alkalinity perturbations under existing regulatory frameworks. Community
engagement and outreach are other areas that will be important to address, especially when the
alkalinity perturbation is large and uncontained. Ideally, local communities should be engaged at
the earliest possible stage since social license to operate is critical for the success of CDR
projects (Nawaz et al., 2022). For a more detailed discussion of legal and social issues see
Steenkamp et al. (2023, this Guide) and Satterfield et al. (2023, this Guide).

With these overarching questions in mind, we discuss considerations for OAE field experiments
in more detail below.

## 2. Research Methods

### 2.1 Types of alkalinity addition

Field experiments of OAE present many challenges. One of the biggest obstacles to success is
tracking alkalinity added to an open system. Methods for adding alkalinity can be divided into
two general approaches: (1) *in situ* or coastal enhanced weathering from the addition of ground
alkaline minerals and rocks with the expectation they will dissolve directly in seawater, and (2)
aqueous alkalinity additions, or the addition of 'pre-dissolved' alkalinity to seawater that can be
generated in numerous ways including through dissolution reactors and electrochemical
techniques (Eisaman et al., 2023, this Guide). Tracking the added alkalinity, and subsequent
CDR, under each approach comes with its own unique set of challenges and considerations.
Adding ground minerals and rocks to an open system presents two distinct scientific challenges.
First, for alkalinity to be considered additional it needs to be attributed to the dissolution of the
solid material. This can be accomplished through a range of techniques including measuring the
loss of mass of the added material or using geochemical tracers in the receiving waters.
Determining dissolution kinetics *in situ* will be particularly important and they are likely to vary
between different deployment environments and strategies (e.g., coastal vs open ocean). For
example, the chemistry (e.g., salinity, pH, temperature) of the waters where the mineral is added
could vary significantly depending on the environment (e.g., beach face, estuary, continental
shelf). Chemical (e.g., seawater conditions such as salinity, $p$CO$_2$, and silica concentrations) and
physical (e.g., grain size and surface area of the added material) will be critical in determining
dissolution rates (Rimstidt et al., 2012; Montserrat et al., 2017; Fuhr et al. 2022). Physical
abrasion through wave action and currents is also likely to be an important control on dissolution
(Flipkens et al., 2023). Field experiments will help translate dissolution kinetics from laboratory
and mesocosm experiments to natural systems, which is not often straightforward due to
complicated biogeochemical processes that are hard to replicate *ex situ* (Morse et al., 2007).
The second major challenge is common to both solid and aqueous approaches and involves
tracking the added alkalinity, which becomes a particularly difficult problem in open-system
field experiments where water is freely exchanged. Depending on the objectives of the field
deployment, this is likely to be a main scientific concern. However, it is important to note that
tracking the added alkalinity does not necessarily equate to observing CDR (i.e., an increase in
seawater CO$_2$ stored as bicarbonate or carbonate). Observing an increase in atmospheric CO$_2$
stored as seawater dissolved inorganic carbon comes with its own set of challenges that are
discussed in depth by Ho et al. (2023, this Guide).

Whether or not the alkalinity is derived from in situ mineral dissolution or direct aqueous
additions, for OAE to be successful atmospheric CO$_2$ needs to be taken up by seawater or CO$_2$
effluxes from seawater to the atmosphere need to be reduced. Therefore, understanding the
physical mixing and air-sea gas exchange dynamics of the deployment site will be a factor of
interest for many field studies. Incorporating physical mixing models with biogeochemical
processes will likely be the end goal of many field experiments focused on MRV (Ho et al. and
Fennel et al., 2023, this Guide). Choosing sites with minimal mixing of different water masses or
with well-defined diffusivities could facilitate tracing released alkalinity and subsequent air-sea
CO$_2$ fluxes. While minimal mixing of different ocean water masses may be desired, higher wind
speeds and wave action will increase the rate of air-sea gas exchange and may make CDR easier
to measure. Background seawater chemistry will also be important in controlling air-sea gas
exchange. For example, sites with naturally lower buffering capacities will see greater changes in
CO$_2$ per unit of added alkalinity (Egleston et al., 2010; Hauck et al., 2016). The release of
conservative tracers will likely be useful for field experiments that wish to track the added
alkalinity and is discussed in more detail below (Section 2.5).

Other experimental considerations related to the type of alkalinity perturbation include the
duration and location of alkalinity addition, which will be important for environmental and
regulatory considerations. Alkalinity can be added once, in timed doses, or continuously.
Aqueous alkalinity could be added directly to seawater, but the rate of this addition will likely be
important, especially for avoiding secondary precipitation (Hartmann et al., 2023; Moras et al.,
2022, Fuhr et al., 2022). Compared to experiments based on one-time additions of aqueous
alkalinity or fast dissolving solid-phase materials (e.g., $Ca(OH)_2$), field experiments adding solid
minerals with comparatively slow dissolution rates (e.g., olivine) will likely need to consider
longer experimental time frames to incorporate the monitoring of mineral dissolution. However,
the timescale of each experiment will ultimately depend on the scientific objectives and could
last from weeks to years and even decades. Location is another important factor that will
influence logistics. For example, amending beach sand with alkaline minerals will present
different challenges compared to the addition of alkaline material to outfalls that discharge into
the ocean. Based on these and other considerations, each field experiment will require specific
spatial and temporal sampling schemes to be developed. These sampling schemes should be
planned well in advance of any perturbation and may require preliminary sampling campaigns to
fine tune.

**2.2 Alkalinity sources**

OAE via coastal enhanced weathering can be accomplished using a variety of naturally occurring
and human-made rocks and minerals (Table 1). The addition of these rocks and minerals is done
after they have been ground to a desired grain size with many unique application techniques
proposed after the initial grinding step (see Eisaman et al., 2023, this Guide). The simplest
application is done via sprinkling the ground material on the ocean surface, although this has
many disadvantages including sinking and advection of the material before it dissolves (Koehler
et al., 2013; Fakharee et al., 2023), although deployment in boat wakes may be viable (Renforth
et al., 2017; He and Tyka, 2023). Other application techniques include spreading material in
coastal ecosystems such as on beaches, marshes, riverbeds, and estuaries, which have the
potential to enhance dissolution through processes such as physical wave action and favorable
water chemistry. However, the complex physical and biogeochemical processes that promote
enhanced weathering in coastal ecosystems can make field experimentation more complicated by
creating strong spatiotemporal modes of variability in water chemistry. To make results more
broadly applicable, field experiments should attempt to mimic real world alkalinity application
scenarios such as those described above.

Any field experiments that add ground material to marine ecosystems may consider tracking the
fate of that material from the addition site. Experiments could also artificially contain the
material using barriers to avoid rapid loss of the ground material via currents, however, this
could make the experiment less comparable to real world OAE deployments. Sampling should
extend from the water column into areas where the material is added including sediments and
pore waters.

Likely environmental impacts associated with coastal enhanced weathering come from the
physical impacts of adding finely ground material or the chemical release of trace elements and
other contaminants. Both processes could have associated risks and/or co-benefits for a range of
ecological processes and biogeochemical cycles (Bach et al., 2019). For example, the addition of
finely ground material could lead to increased turbidity from the initial addition, subsequent
resuspension, or secondary precipitation of particulates in the water column. Additionally, any
release of nutrients or heavy metals from the dissolving material could alter primary production
or cause harm to biological systems. The bioaccumulation of toxic metals in higher trophic level
organisms, especially those of commercial importance, is a widespread concern.

Safety criteria should be put in place that can create a pause in the field experiment or prevent
future experiments of the same type from taking place. These guardrails should be developed by
the broader OAE community but may include obvious damage or health impacts to ecologically
important organisms such as primary producers and keystone species, large and unexpected
changes in biogeochemical cycles, and the general deterioration of environmental conditions.
Risk-benefit analysis may be particularly useful in determining whether projects can or should
move forward and may already be included in regulatory requirements through existing
frameworks such as environmental impact assessments.

Aqueous and slurry-based additions of alkalinity provide different benefits and challenges
compared to solid forms of alkalinity feedstock. One of the primary benefits of aqueous
additions is that the alkalinity has been pre-dissolved, avoiding the often slow dissolution
kinetics of minerals and rocks in seawater. Aqueous alkalinity can be generated by two main
mechanisms (1) the dissolution of alkaline rocks and minerals in reactors, and (2)
electrochemical processes that generate alkalinity by splitting seawater or other brine streams
into an acid and base (Eisaman et al., 2023, this Guide). For some materials, such as $Ca(OH)_2$
and $Mg(OH)_2$, dissolution slurries are formed and a combination of particulate and aqueous
alkalinity can be dosed into seawater. Any particulates that are dosed from the slurry need to
dissolve, meaning dissolution kinetics in seawater will be critical. However, the dissolution of
these materials tends to be much quicker than with rocks and minerals (Table 1). There are
important processes that need to be considered when adding aqueous alkalinity, including the
unintended precipitation of calcium carbonates due to locally elevated saturation states
(Hartmann et al., 2023; Moras et al., 2023).

Field experiments that use aqueous or slurry-based alkalinity additions will need to assess the
impacts on seawater chemistry at the source of addition and across a dilution radius. Depending
on the type of experiment and magnitude of additions this dilution radius could extend upwards
of kilometers, but the magnitude of the perturbation to carbonate chemistry would become
smaller the further away from the alkalinity source (He & Tyka, 2023). The potential
environmental impacts from aqueous type alkalinity additions will be similar to those discussed
for coastal enhanced weathering, but also include extreme localized changes in carbonate
chemistry.

**Table 1.** Types of alkalinity sources and considerations for each.

| Alkalinity Source | Solid/Aqueous | Dissolution kinetics | Dissolution co-products |
|---|---|---|---|
| NaOH | Aqueous | Instantaneous but can | Alkalinity, $Na^+$ |

| | | induce brucite $(Mg(OH)_2)$ precipitation when NaOH elevates pH >9. Brucite re-dissolves relatively quickly in most cases. | |
|---|---|---|---|
| Manufactured and natural Mg derived alkalinity sources (e.g.,, brucite) | Solid or aqueous slurry | Relatively fast but a combination of dissolution rates both in the receiving and dosing waters. | Alkalinity, limited amounts of nutrients and trace metals (generally less than silicates), $Mg^{2+}$. |
| Silicates (e.g. olivine, basalt, wollastonite) | Solid | Relatively slow dissolution kinetics, but rates are different between silicates. | Alkalinity, silicate, trace metals. Materials need to be individually assessed prior to their use. |
| Manufactured lime-derived alkalinity sources (e.g. quicklime, ikaite)) | Solid or aqueous slurry | Relatively fast but different kinetics between lime products. | Alkalinity, limited amounts of nutrients and trace metals (generally less than silicates), $Ca^{2+}$. Materials need to be individually assessed prior to their use. |
| Iron and steel slag | Solid | Components within steel slag that provide alkalinity (e.g. CaO) dissolve relatively fast but different iron and steel slag contain different amounts. | Alkalinity, $Ca^{2+}$, $Mg^{2+}$, silicate, phosphate, and trace metals. Materials need to be individually assessed prior to their use. |
| Natural and synthetic carbonates (e.g. calcite, aragonite) | Solid | They don't dissolve under common surface ocean carbonate chemistry conditions. Dissolution rates can be higher in microenvironments such as corrosive sediment porewaters | Alkalinity, phosphate in some mined sources, dissolved inorganic carbon. |

| | | where saturation is low due to respiratory $CO_2$. | |
|---|---|---|---|
| | | | |


## 2.3 Considerations for site selection

Careful consideration should be given to site selection and experimental design to make sure the
study adequately address the specific research questions and goals. Some aspects of the field site
that will be important include ecosystem- and site-specific characteristics, the prevailing
meteorological and oceanographic conditions, and natural spatiotemporal variability. Logistical
considerations for site selection include physical access, permitting, availability of electricity,
ship time, and consideration of the local community. These considerations will grow with the
scale of field experiments and will likely be first-order determinants of where field experiments
take place. For example, proximity to a marine institute (for land-based approaches) or access to
a research cruise (for open ocean approaches) may be desirable. Logistics will ultimately
determine where operational OAE deployments take place and early field experiments will help
to elucidate important issues including the impacts of life cycle emissions on CDR.
OAE field experimentation requires careful assessment of the field site prior to alkalinity
additions to provide foundational knowledge of the site characteristics. Scientific considerations
for site selection can be broken down into three categories, the (1) physical, (2) chemical, and (3)
biological properties of each site. Important considerations for each category are provided in Box
1. To facilitate baseline assessments and site selection we propose Table 2 as guidance for
relevant parameters to measure. We note that this list is broad, however it is not exhaustive and
specific field sites may require the monitoring of different or additional parameters. Furthermore,
some of the listed parameters may be more applicable to specific OAE approaches. Preliminary
knowledge of the field site will inform both the experimental design as well as interpretation of
data and experimental outcomes. Due to the large investments in cost and time required to collect
baseline data, locations with a wealth of pre-existing scientific data may be considered. This
baseline data could be available in the peer-reviewed literature and/or from publicly available
coastal and open ocean time-series (e.g., Sutton et al., 2019).
**Box 1.** Scientific considerations for field experiments.

**Physics:**
- What are the expected dilution rates of the added alkalinity?
- What is the site turbulence and how will this impact alkalinity additions (e.g., keeping particles in suspension)?
- What is the natural light penetration and what impacts could increases in turbidity have on this?
- What is the residence time of water in the surface ocean or mixed layer and how does this relate to the estimated air-sea equilibration time?
- What is driving air-sea gas exchange?
- Will changes in turbidity impact the albedo of the experimental site?
- What is the potential for the lateral export and exchange of alkalinity and other materials?
- Is there the potential for physical disturbance (e.g. impacts of alkalinity additions on physical water mass parameters such as density or the physical impacts of adding undissolved minerals to the benthos)?
- Where will the alkalinity signal be most observable (e.g., pore water vs. water column)?

**Chemistry:**
- What are the natural carbonate chemistry conditions?
- What modes of variability (e.g., daily, seasonal, interannual) impact seawater chemistry?
- How will variations in seawater chemistry impact signal to noise?
- How will seawater chemistry impact mineral dissolution rates?
- Is there potential to disturb the natural concentrations of macro- or micronutrients or toxic metals through dissolution by-products?
- How do anthropogenic sources of alkalinity interact with (and potentially modify) natural sources and sinks of alkalinity?

**Biology:**
- What organisms (benthic and pelagic) are present in the study area and what are their relative sensitivities to fluctuations in seawater carbonate chemistry (if known)?
- Are there culturally or commercially important species present?
- Are there endangered or rare species present? Is the site a nursery and/or nesting ground? Are there keystone species and/or important primary producers present? These considerations will likely be part of the permitting process.
- Are there times of the day or seasons with elevated species or ecosystem sensitivities?
- What are the trophic dynamics in the environment, and how might the food web be impacted (e.g., shifts in predator/prey relationships)? What are the cascading implications for the ecosystem as a whole? Might effects be transferred beyond the study site via migratory species?
- Could particulates (e.g., ground rock) cause physical damage prior to dissolution?

**Table 2.** Parameters that could be considered in assessing sites for OAE field experiments. Importantly, some parameters summarized below may require a baseline assessment over sufficiently long time frames to cover the intrinsic variability of physical, chemical, and biological parameters in the studied system. For example, baseline assessment of marine food web structure will likely require a prolonged monitoring effort before (and after) the OAE deployment to have a higher chance of detecting OAE-induced effects on marine biota.

| Parameter | Rationale | Potential pathway for assessment |
|---|---|---|
| Dilution rate | - Exposure risk to alkalinity and mineral dissolution products.<br>- Detectability of OAE-induced chemical changes. | Tracer release experiment (section 2.5). |
| Turbulence | - Physical energy input to keep ground particles near the sea surface during dissolution. | Microstructure profiler. |
| Residence time of perturbed patch in surface ocean | - Determination of residence time of an OAE-perturbed patch in the surface to assess whether there is enough time for air-sea equilibration with the atmosphere. | Risk assessment for incomplete air-sea $CO_2$ exchange (He and Tyka, 2023; Bach et al., 2023). |
| Transboundary transport | - Determination of whether there is a high risk for OAE-derived chemicals to be transported into sensitive areas (e.g. marine protected areas, other state territories) in high concentrations. May be useful for residence time as well. | - Tracer release experiment<br>- Virtual Lagrangian particle tracking.<br>- Utilizing natural tracers observable via remote sensing (e.g., CDOM or Gelbstoff).<br>- Mixed layer depth. |
| Light penetration | - Determination of light environment to assess to what extent the addition of particulate alkalinity source could impact turbidity. | Light loggers, turbidity, CTD casts. |
| Carbonate chemistry conditions | - Baseline of mean conditions and variability to assess how much change OAE must induce to become detectable.<br>- Determination if OAE-related changes are likely to affect marine organisms. | Dickson et al. (2007) and ocean acidification literature. Schulz et al., (2023, this Guide) |

| | | |
|---|---|---|
| Macronutrients | - Assessment of whether the designated system is prone to macronutrient fertilization via OAE. (Note that not all OAE approaches would introduce macronutrients into the ocean system). | Standard photometric approaches (Hansen and Koroleff, 1999). Experimental assessment of limiting elements. |
| Micronutrients | - Assessment of whether the designated system is prone to micronutrient fertilization via OAE. (Note that not all OAE approaches would introduce micronutrients into the ocean system). | GEOTRACES cookbook. (https://www.geotraces.org/methods-cookbook/) Experimental assessment of limiting elements. |
| Marine food web structure | - Assessment of the planktonic and/or benthic food web structure prior to testing an OAE deployment. | There is a whole range of surveying tools that could be applied depending on the size and abundance of organisms. Applied methods could range from OMICS (including eDNA), optical observations, acoustics, and flow cytometry. |
| Risk of damaging organisms by adding ground minerals | - Providing knowledge of whether organisms could be physically harmed, for example through covering them with mineral powder. | Same range of methods as for the food web assessment. |
| Endangered species | - Clarification if endangered species could be present at the designated field site. | Same range of methods as for the food web assessment. Plane or drone surveys can help to confirm sightings of larger organisms and there may be online resources to be utilized (e.g., WhaleMap). Furthermore, local knowledge should be sought after from the diverse range of stakeholder groups. For example, consultation with indigenous communities, fishermen, local authorities, and environmental agencies. |
| Foraging/breeding ground | - Clarification if the designated field site is an important | Same range as for endangered species assessments. |

| | breeding/foraging area for migratory organisms. | |
|---|---|---|


## 2.4. Measurement considerations

What to measure and the type of instrumentation needed will ultimately depend on the site, scale,
and goals of each individual experiment and should be considered on a case-by-case basis. For
example, depending on the alkalinity source utilized (Table 1), it may (e.g., in the case of
olivine) or may not (e.g., in the case of NaOH) be a priority to measure trace metal or nutrient
concentrations. In addition to alkalinity type, the experimental scale will also dictate
measurement considerations. For example, if the scale of the perturbation is small or the signal is
very dilute, environmental impacts will not likely be measurable far from where the perturbation
takes place. If there is a large addition of alkalinity, especially in a semi-enclosed system, both
environmental impacts and changes in chemistry will be easier to detect. Ultimately, when OAE
is done at a larger scale (e.g., millions of moles alkalinity) it is likely that large changes in
seawater chemistry will wish to be avoided to reduce environmental impacts and avoid
secondary precipitation. This presents an interesting challenge to conducting field experiments,
as the dilution of alkalinity and ultimately $CO_2$ signal will make MRV more challenging (Ho et
al., 2023, this Guide).
Seawater carbonate chemistry measurements will be central to most sampling schemes. To cover
the appropriate spatial and temporal scales, traditional bottle sampling will likely have to be
combined with state of the art *in situ* sensors (Bushinsky et al., 2019; Briggs et al., 2020; Ho et
al., this Guide). Bushinsky et al. (Figure 1; 2019) provides a comprehensive overview of the
spatiotemporal capabilities of existing carbonate chemistry sensors and platforms, and care
should be taken to make sure sensors are appropriate for measurements in seawater. The
appropriate methods and protocols for sampling and analysis are outlined in other chapters in this
guide (Schulz et al., 2023, this Guide) and in the Guide to Best Practices (Dickson et al., 2007).
Some general considerations for field experiments include appropriately characterizing the
natural variability that occurs at the field site through space and time. While total alkalinity
titrations should remain a priority, at least two carbonate chemistry parameters (e.g., total
alkalinity, dissolved inorganic carbon, pH, or $pCO_2$) should be measured for each sample. It is
important to note that the combination of $pCO_2$ and pH is not ideal when calculating $CO_2$
chemistry (e.g., using CO2SYS) due to the elevated errors when combining those parameters in
determining the rest of the carbonate chemistry system in seawater (Lee and Millero, 1995).
Currently, commercially available autonomous sensors exist for pH and $pCO_2$, with sensors in
development for both TA and DIC (Fassbender et al., 2015; Briggs et al., 2020; Qiu et al., 2023).
While autonomous sensors generally have greater uncertainty than bottle samples coupled with
laboratory analysis, they will likely play an important role in sampling schemes to help cover
adequate spatial and temporal resolution in naturally variable marine systems.
While monitoring the background variability and subsequent additions of alkalinity will be
critical, scientists may also wish to directly measure fluxes of carbon at the field study site (Ho et
al., this Guide). The direct measurement of carbon fluxes can be accomplished via different
methods including benthic and floating chambers, eddy covariance and other benthic boundary
layer techniques, and mass balances. These techniques have benefits and drawbacks, including
having to enclose the natural system (e.g., chambers) and elevated uncertainty that could be
outside of the expected changes due to the perturbation (e.g., eddy covariance). Benthic chamber
measurements may be particularly important to quantify the dissolution of minerals and rocks
added to sediments. Ultimately, any measurements of fluxes due to OAE activities will likely
need to be coupled with numerical modeling to estimate the overall drawdown of atmospheric
$CO_2$ (Fennel et al., 2023, this Guide).
Field experiments should be informed by other scientific studies as much as possible (e.g.,
studies based on laboratory experiments, mesocosm studies, natural analogs, and numerical
modelling). While not necessarily directly translatable to natural systems (Edmunds et al., 2016;
Page et al., 2021), these types of studies can provide first order assessments on safety and
efficacy, helping to prevent unintended harmful ecological side effects when conducting large
scale perturbations.
Other measurements that may be useful during OAE field experiments are outlined in Table 2. It
is important to note that this list is not meant to be exhaustive, and measurement selection will
have to be made on a case-by-case basis. Considering the difficulties of tracking water masses in
an open system, the next section is a more detailed discussion on tracers for monitoring mixing
and dilution of water within the OAE field experiment site. Tracking added alkalinity will be
critical to determine the impacts and efficacy of alkalinity enrichments and may be one of the
biggest challenges facing OAE field experiments.
**2.5 Dual tracer regression technique**
If the goal is to track alkalinity additions and measure their effects on carbon fluxes (i.e., net
ecosystem production or air-sea exchange), a dual tracer regression method can be used (e.g.,
Albright et al. 2016 & 2018). This approach uses the change in ratios between an active tracer
(alkalinity) and a passive tracer (dye, artificial gas tracer, Table 3) to assess the fraction of added
alkalinity taken up or released by biogeochemical processes in the system. Passive tracers do not
affect fluid dynamics and are passively advected by the surrounding flow field. The use of
passive tracers, such as dye tracers (e.g., rhodamine, fluorescein) or artificial gas tracers (e.g.,
$SF_6$, $CF_3SF_5$) that do not occur in nature helps eliminate background noise. Additional
considerations include how many tracers to use and what information each tracer provides (Table
342 3).


During a dual tracer experiment, changes in the active tracer (alkalinity) result from mixing,
dilution, and biogeochemical activity, whereas changes in the passive tracer are due solely to
mixing and dilution. By comparing the alkalinity to dye ratios before (e.g., upstream) and after
(e.g., downstream) the water mass interacts with a study area, it is possible to isolate the change
in alkalinity that is due to biogeochemical processes such as calcium carbonate precipitation and
dissolution (Figures 1 & 2). This technique is an extension of Friedlander et al. (1986) and may
have applications in other areas of research pertinent to marine CDR, such as nutrient or
pollution assessments and the uptake of industrial or agricultural waste. The primary
experimental criteria for the dual tracer technique are that the active and passive tracers are
added in a fixed ratio and at a fixed rate, in areas where there is a dominant flow direction,
dispersion or dilution.

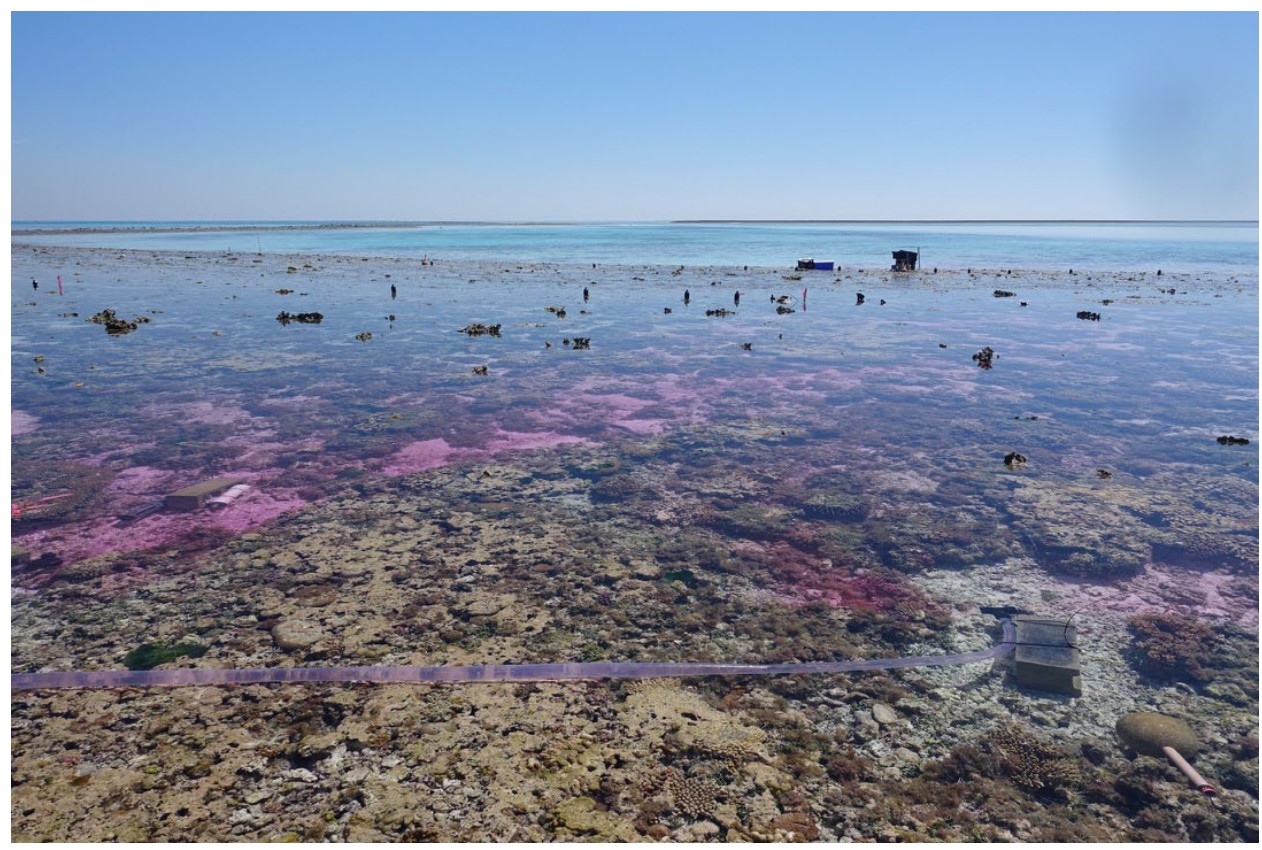

**Figure 1.** Rhodamine dye flowing over a coral reef flat study site during a study in One Tree
Island, Australia (Albright et al. 2016). NaOH was used as an active tracer to raise alkalinity, and
rhodamine was used as a passive tracer to account for mixing and dilution. Changes in the
alkalinity-dye ratios were used to isolate the change in alkalinity flux that was associated with an
increase in net community calcification on the reef flat.

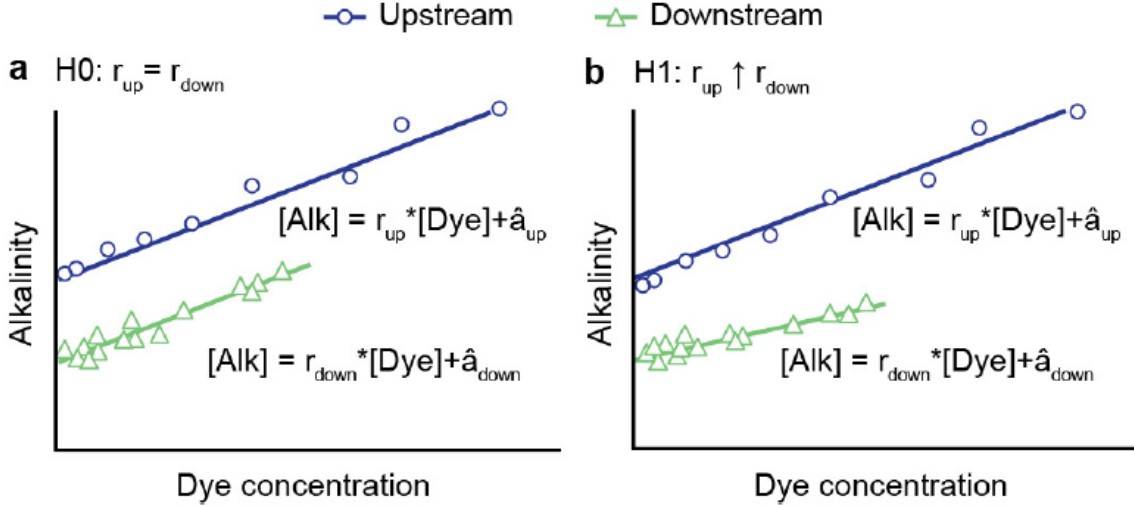

**Figure 2.** Theoretical representations of the null (H0) and alternative (H1) hypotheses for a dual
tracer regression experiment where NaOH was used as a source of alkalinity and rhodamine dye
was used as a passive tracer (from Albright et al. 2016). (a). In H0, the benthic community does
not take up added alkalinity. Here, the change in alkalinity between the upstream and
downstream transects would not be systematically related to the dye concentration, and the ratio
of the alkalinity–dye relationship, r, would not be expected to change between the upstream and
downstream locations (that is, $r_{up} = r_{down}$). (b). In H1, an uptake of added alkalinity occurs by the
benthic community. Here, areas with more alkalinity (and more dye) change at a different rate
than areas with less alkalinity (and less dye), resulting in a change in the alkalinity–dye slope
(that is, $r_{up} > r_{down}$).

**Table 3.** Passive tracers that are available and commonly used for use in field experiments and
considerations for each. Additional tracers may be useful that are not listed in this table,
including Helium 3 and Tritium.

| Tracer | Type | Pros | Limitations | Lifespan |
|---|---|---|---|---|
| Rhodamine | Fluorescent dye | Sensor-based, high frequency (>4 Hz) detection, platform flexibility, detection from space and/or the sky for surface releases. | Optically degrades and absorbs to particles, not good for longer-term studies, not as good signal to noise/detection limits as inert gas tracers. | Several days to weeks |
| Fluorescein | Fluorescent dye | Sensor-based, high frequency (>4 Hz) detection, platform flexibility, detection from space and/or the sky for surface releases. | Degrades optically - not good for longer-term studies (>24h) | <24 h |
| SF6 | Artificial gas | Inert; capable of being measured at very low concentrations; able to quantify mixing and residence time; good for large-scale ocean tracer release experiments. | Lower frequency detection and less flexibility with platforms, requires discrete measurement. High global warming potential. | years |
| Trifluorom ethyl sulfur pentafluori | Artificial gas | Good for large-scale ocean experiments. | Difficult to obtain, lower frequency detection and less flexibility with | years |

| de (CF₃SF₅) | | | autonomous platforms, requires discrete measurement. High global warming potential. | |
|---|---|---|---|---|

**2.6 Detecting change and the importance of controlled experiments**

Separating an experimental 'signal' from the background 'noise' inherent in natural systems can be challenging, especially in field experiments where replication may not be practical (Carpenter, 1990). Gaining baseline knowledge on the physical, chemical, and biological components of the study site should be a priority. There is often considerable natural variability in marine systems, and especially in coastal systems, due to fluctuations in biological activity, hydrodynamics, seasonal and/or interannual influences, and others (Bates et al., 1998; Bates 2002; Hagens and Middelburg, 2016; Landschützer et al., 2018; Sutton et al., 2019; Kapsenberg and Cyronak, 2019; Torres et al., 2021). Fully characterizing this variability could take many years, which may create significant barriers to experimental progress in the field. Therefore, we recommend that any potential modes of spatiotemporal variability be recognized and evaluated while planning field experiments. For instance, in coastal systems with river and groundwater inputs it will be important to know the impact that freshwater has on carbonate chemistry.

Where possible, conducting controlled experiments will help to maximize the ratio of signal to noise, thereby improving statistical power to detect experimental effects. The pros and cons of replicating experimental controls in space versus time should be taken into consideration. For many field experiments (and natural analogs; see Subhas et al., 2023, this Guide), sample size will be inherently limited (e.g., one, or few study sites); therefore, conducting controls in time (e.g., every third day) may be the best option. For studies with limited (or no) replication, there are statistical methods that can be used to isolate effects pre- and post-treatment (Carpenter, 1990). Numerical simulations and machine learning based network design are potentially valuable tools to optimize observational networks to detect experimental change.

**3. Additional considerations**

***Permitting.*** Addressing regulatory requirements is critical prior to conducting field experiments.
The spatial and temporal scale of the field trial, as well as the specific considerations of the
deployment site (e.g., protection status) will determine permitting requirements. Engaging with
this process early is advised - for example, understanding who the permit-granting authorities are
for a given area and timelines for associated regulatory processes. In some cases, the use of
existing infrastructure (e.g., wastewater discharge sites) and environmental projects (e.g., beach
renourishment) may offer ways to streamline experiments, although permitting will be governed
by existing regulations. For a detailed discussion on legal considerations, see Steenkamp and
Webb (2023, this Guide).
***Community engagement and social considerations of field experiments.*** The likelihood of
harmful ecological consequences from OAE field experiments remains unclear and will
ultimately depend on the technology and temporal and spatial scale of the experiment. Field
experiments evaluating CDR approaches carry the risk of unintended consequences and impacts
over large spatial scales, so appropriate scaling (e.g., starting small) is necessary (NASEM,
2022). In response to these unknowns, researchers should follow the key components for a code
of conduct for marine CDR research, e.g., as outlined by Loomis et al. (2022), which details best
practices that encourage responsible research amongst both the public and private sectors.

Social license to operate is critical for the success of CDR projects and researchers have an
obligation to involve the full community of people (public and stakeholders) who may be
impacted by the research (Nawaz et al., 2022; Cooley et al., 2023). Therefore, public outreach is
important both before and during field experimentation. The study site will determine the
potential for community engagement. Coordinating with local and/or regional organizations who
are connected to relevant stakeholders (for example, your local SeaGrant office if in the United
States) will be helpful. For additional discussion on social considerations of OAE field trials, see
Satterfield et al. (2023, this Guide).
***Collaboration and data/information sharing.*** Considering the inherent challenges to OAE field
experiments (cost, permitting, access, logistics, environmental safety), fostering interdisciplinary
and collaborative teams will help ensure the greatest return on investment. Examples of ways to
foster collaboration include, developing test-bed field sites that are open to participation from
diverse groups, making efforts to include groups who may not traditionally have access to and/or
the capacity for field campaigns, and including travel support in grant applications to support
external collaborators. Making concerted efforts to share information, resources, and ideas will
allow researchers to combine knowledge and resources in ways that might not have been
possible when working alone, thereby advancing OAE technology and science at a faster pace.
When publishing in peer-reviewed literature, uploading data to publicly available data
repositories and publishing in open access journals following best practices should be prioritized
(Jiang et al., 2023, this Guide).
Inclusivity and transparency during OAE field trials are crucial to ensure that knowledge gained
is fed back into scientific and other communities efficiently, iteratively informing and refining
the next generation of experiments. Some field experiments will mimic plans for real world OAE
deployments and should therefore be done in collaboration with relevant stakeholders across
science, industry, policy, and communities. To foster collaboration and technology transfer, we
advocate for a centralized platform and/or organization to share data and information in this
rapidly evolving field. This might look like a centralized, freely accessible platform for early
and/or 'real-time' information sharing (i.e., before publication) that can facilitate faster
information exchange within the research community (e.g., data sharing, permitting issues). Two
existing options that could help fill this gap are the OA information exchange
(https://www.oainfoexchange.org/index.html) and the Ocean Visions community
(https://community.oceanvisions.org/dashboard). It may prove useful to designate core working
groups of experts in various aspects of CDR that investigate specific needs and priorities and
work to synthesize and share existing knowledge in the context of field experiments. This
approach has been adopted by other scientific disciplines in high priority, rapidly evolving, and
highly collaborative fields, greatly benefiting the scientific community at large (e.g., the Coral
Restoration Consortium, https://www.crc.world/ - and associated working groups). Coordinating
field trials with research groups conducting laboratory and mesocosm experiments, studying
natural analogs, and undertaking modeling efforts will help strengthen the interpretation and
extrapolation of results.
**4.1 Conclusion and Recommendations**
Given that few OAE field studies have been conducted to date, there is much to learn from the
earliest experiments with respect to experimental design, measurement and monitoring,
deployment considerations, environmental impact, and more. Early experiments will only engage
with a fraction of the temporal and spatial scales involved in full-scale operational OAE, and
longer-term and larger-scale studies will become increasingly important to reveal scale-
dependencies as the field develops. It is important that marine CDR research is hypothesis-
driven, structured, deliberate, and well-planned to best inform future decision-making about
OAE techniques and deployments. Careful consideration of the physical, chemical, and
biological components of the study area will help inform the experimental approach. The use of
baseline studies (both previous and contemporary to the OAE deployment) and controls will help
to maximize signal-to-noise ratios and identify experimental effects. The timescale of OAE field
experiments should not be underestimated, especially when considering permitting and the data
needed to capture the baseline variability in natural systems.
Considering the urgent timeline required for humanity to meet our climate goals, field
experiments need to move forward swiftly yet deliberately. To ensure the success of OAE,
diverse perspectives from research, industry, policy, and society must converge, demanding
transdisciplinary thinking and a commitment to open and transparent science. Central to this
ambitious undertaking are the early field experiments, results from which will ultimately
determine the successes and failures of OAE projects and technologies.
**4.2 Key recommendations for field experiments relevant to the research of ocean alkalinity**
**enhancement**
1. Ensure inclusivity and transparency (community engagement, data sharing, etc) for

485       OAE field experiments to both advance the field as quickly as possible, and to ensure

486       the field progresses in a socially responsible manner.

2. Assess the potential risks and benefits for any perturbation. Proceed according to the

488       code of conduct and precautionary principles.

3. Develop methods to track signal versus noise in highly variable environments,

490       including robust baseline studies to characterize underlying variability (biological,

491       chemical, physical), and the inclusion of controlled experiments such as chamber

incubations to isolate treatment effects.

493         4.   Consider logistical constraints and opportunities.


**Competing Interests**

The contact author has declared that none of the authors has any competing interests.

**Disclosure Statement**

LTB is scientific advisor to Submarine, a start-up service provider for monitoring, reporting, and
verification of marine CDR. TC is an advisor on the Carbon-to-Sea Initiative OAE Field Site
Steering Committee.


**Acknowledgements**

This is a contribution to the "Guide for Best Practices on Ocean Alkalinity Enhancement
Research". We thank our funders the ClimateWorks Foundation and the Prince Albert II of
Monaco Foundation. Thanks are also due to the Villefranche Oceanographic Laboratory for
supporting the lead authors' meeting in January 2023. LTB was supported by the Australian
Research Council through Future Fellowship (FT200100846) and by the Carbon-to-Sea
Initiative.

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

and Extreme Diurnal Variability of Ocean $CO_2$ System Variables Across Marine Environments,
Geophysical Research Letters, 48.