# Peer review of "Field experiments in ocean alkalinity enhancement research Tyler Cyronak1\*, Rebecca Albright2\*, Lennart T. Bach3 1Georgia Southern University, Institute for Coastal Plain Science, Statesboro, GA 2California Academy of Sciences, San Francisco,"

_State of the Planet, 2023_

## Author Response (AR1)

**Reviewer 1:**

**Overview**

This chapter is well-written and I have no major objections to it. My main criticism would be that it is fairly light on concrete recommendations, examples and references. I understand that this is in large part due to the infancy of this research area and the very limited number of field experiments that have been performed/published. Nonetheless, perhaps the authors could be more specific in their recommendations by utilizing related literature. For example, can the many field experiments of mCDR based on iron fertilization offer specific insights into OAE best practices?

We appreciate the reviewer's constructive comments and suggestions. Based on these and other comments, we have tried to incorporate more concrete and specific recommendations throughout the manuscript.

Given that the principal goal of OAE is CDR the chapter could do with more focus on air-sea gas exchange. I would argue that the principal objective of OAE is CDR (although minimizing OA impacts on organisms may be a secondary consideration). With this in mind, perhaps the authors could do more to highlight the types of oceanographic and carbonate chemistry conditions/measurements that might measurable CDR.

We agree that measuring oceanic CO2 uptake via air-sea gas exchange following alkalinity enhancement is one of the many important considerations to determine the success of an OAE operation. Since Ho et al. (this issue) provides a detailed discussion of the topic we have opted to not expand our discussion on it but refer to this chapter throughout our manuscript.

**Specific comments**

L62-64. Will projects adding ground alkalinity "require" tracking of mineral dissolution and the fate of alkalinity? Although these are interesting related research questions, I'm not convinced it's a prerequisite to tracking CDR which in most cases is likely to be the principal objective.

Based on multiple reviewers comments we have toned down our language when we previously used words like 'require'.

In principle we agree that air-sea CO2 influx (we assume this is what is meant by CDR here) could be sufficient if the influx could be linked to a seawater CO2 deficit generated through OAE. Establishing such links between anthropogenic alkalinity and air-sea CO2 flux is possible in models (He and Tyka, 2023); however, it will be close to impossible in reality because our observational capacity at the necessary spatio-temporal scales is insufficient (Bach et al., 2023). As such, it is almost certain that satisfying MRV will require monitoring of different processes of an OAE operation. Generating alkalinity (e.g., via dissolution of alkaline minerals) is one of these crucial steps, and we are confident that this process requires specific investigation.

Indeed, currently evolving MRV protocols associated with field experiments investigate alkalinity generation via dissolution in great detail.

L 84-86 See previous point on the focus on alkalinity tracking as opposed to CDR. The rationale behind tracking alkalinity could perhaps be better introduced. I consider alkalinity tracking more important to assessing environmental impacts/co-benefits than CDR.

See above comment.

L99-103. Perhaps missing something on physical dynamics here and there importance in relation to CDR (e.g. ocean kinetic energy, mixed layer depth, waves, tides, wind speeds).

The text was amended to include this with an added reference:

"Chemical (e.g., seawater conditions such as salinity, $pCO_2$, and silica concentrations) and physical (e.g., grain size and surface area of the added material) monitoring will be critical in determining dissolution rates (Rimstidt et al., 2012; Montserrat et al., 207; Fuhr et al. 2022). Physical abrasion through wave action and currents is also likely to be an important control on dissolution kinetics (Flipkens et al., 2023). Initial field experiments will help translate results from laboratory and mesocosm experiments on dissolution kinetics to natural systems."

L104. "drawdown of atmospheric CO2" – OAE could also be deployed to prevent natural carbon degassing (e.g. in upwelling regions). Maybe this should be specified. The overall outcome is the same (an increase in the ocean DIC inventory).

Language was changed in the text:

L128: "The second major challenge is common to both solid and aqueous approaches and involves tracking the added alkalinity, which becomes a particularly difficult problem in open-system field experiments where water is freely exchanged. Depending on the objectives of the field deployment, this is likely to be a main scientific concern. However, it is important to note that tracking the added alkalinity does not necessarily equate to observing CDR, or an increase in seawater CO2 stored as bicarbonate or carbonate. Observing an increase in atmospheric CO2 stored as seawater dissolved inorganic carbon comes with its own set of challenges that are discussed in depth by Ho et al. (2023, this Guide)."

L108-109. I wouldn't refer to a "DIC deficit". The deficit that results from OAE is in pCO2 and it's this that equilibrates with the atmosphere.

This point was clarified in the text:

L137: "Whether or not the alkalinity is derived from in situ mineral dissolution or direct aqueous additions, for OAE to be successful atmospheric CO2 needs to be taken up by seawater or CO2 effluxes from seawater to the atmosphere need to be reduced."

L111-112. "minimize mixing" is a bit ambiguous here. While minimal mixing of different ocean water masses may be desired, "mixing" of the surface ocean by higher wind speeds/wave action will increase the rate of air-sea gas exchange and may make CDR easier to measure. Another point that could be made here, or in the signal-to-noise section, is that sites with lower buffer capacity may also better enable measurable CDR as there is greater change in CO2(aq) per unit change in alkalinity (Egleston et al., 2010; Hauck et al., 2016).

We have amended the text to address this comment:

L139: "Therefore, understanding the physical mixing and air-sea gas exchange dynamics of the deployment site will be a factor of interest for many field studies. Incorporating physical mixing models with biogeochemical processes will likely be the end goal of many field experiments focused on MRV (Ho et al, and Fennel et al., 2023, this Guide). Choosing sites with minimal mixing of different water masses or with well-defined diffusivities could facilitate tracing released alkalinity and subsequent air-sea CO2 flux. While minimal mixing of different ocean water masses may be desired, higher wind speeds and wave action will increase the rate of air-sea gas exchange and may make CDR easier to measure. Background seawater chemistry will also be important in controlling air-sea gas exchange. For example, sites with naturally lower buffering capacities will see greater changes in CO2 per unit of added alkalinity (Egleston et al., 2010; Hauck et al., 2016). The release of conservative tracers will likely be useful for field experiments that wish to track the added alkalinity and is discussed in more detail below (Section 2.5)."

L126-127. Is this because of the dissolution timescale? I would expect the limiting factor to be the timescale of air-sea gas exchange not the dissolution timescale of minerals. Perhaps the authors could put broad ranges on these numbers.

The timescale really depends on the objectives of each experiment. However, enhanced weathering will likely need to be monitored for longer periods as dissolution of the minerals will take years to decades. It is difficult to put exact numbers on each field experiment without knowing the exact goals, and we try to clarify this in the text:

L158: "Compared to experiments based on one-time additions of aqueous alkalinity or fast dissolving solid-phase materials (e.g. Ca(OH)2), field experiments adding solid minerals with comparatively slow dissolution rates (e.g. olivine) will likely need to consider longer experimental time frames to incorporate the monitoring of mineral dissolution. However, the timescale of each experiment will ultimately depend on the scientific objectives and could last from weeks to years and even decades."

L139. Deployment in boat wakes to maximize dissolution have also been proposed (Renforth and Henderson, 2017; He and Tyka, 2023).

Added to text:

L176: "The simplest application is done via sprinkling the ground material on the ocean surface, although this has many disadvantages including sinking and advection of the material before it

dissolves (Koehler et al., 2013; Fakharee et al., 2023), although deployment in boat wakes may be viable (Renforth et al., 2017; He and Tyka, 2023).

L154. If the turbidity may be affected it's not impossible that albedo might also be. I think it would be good practice in OAE tests to monitor for any change in ocean surface albedo. The ultimate aim of CDR is to affect the Earths radiative budget and limit anthropogenic warming, unintended impacts on the radiative budget through albedo changes are an uncertainty that has received limited attention.

While this is an interesting point, we feel that it is a bit outside of the scope of this chapter for an OAE specific guide and have opted not to include it in the text. However, we do now mention it in Box 1 (see specific comment below).

Table 1. More detail on the potential trace metals in some of these alkalinity sources would be useful. Should phosphate be listed as a dissolution product of carbonates?

Depending on the source of the carbonates, phosphate can present, and we have clarified this. There is such a wide variety of trace metals within various materials that are classified as the same material (e.g., olivine) depending on where they are mined. Therefore, we have opted to leave the table as is with the statement "Materials need to be individually assessed prior to their use."

L188-190. Prevailing meteorological and oceanographic conditions may also be a consideration.

We have updated the text:

L239: "Careful consideration should be given to the site selection and experimental design to adequately address the specific research questions. Some aspects of the field site that will be important include ecosystem- and site-specific characteristics, the prevailing meteorological and oceanographic conditions, and natural spatiotemporal variability."

Box 1. Physics

- I'd mention potential albedo impacts here.
- I'd split point 4 into 2 points both of which are important. 1) what is the air-sea gas exchange prior to OAE? (I wouldn't call this "natural" as it may be strongly influenced by the anthropogenic carbon) 2. Water is the residence time of water in the surface ocean/mixed layer and how does this relate to the estimated equilibration time?
- Not sure what physical "risks" are being referred to here. Is this referring to local water residence times and the potential impact on MRV?

Chemistry

- The authors could distinguish mean state carbonate chemistry conditions and their variability (diurnal/seasonal/interannual). Both of these will impact signal-to noise.

Biology

- Are there times of the day or seasons with enhanced species/ecosystem sensitivities?

We have updated the text in Box 1 to reflect these comments.

Table 2.

- Some of the transboundary transport tools could be equally used for assessing surface ocean residence times
- Mixed layer depth and atmospheric wind speeds may also be important parameters as they have been in iron fertilization field experiments.
- Ocean biogeochemical models are also likely to be provide useful assessments of water residence time, gas exchange and potential nutrient/primary production impacts.

We have updated the text in Table 2 to reflect these comments.

Section 2.5. Could this section be expanded to dual tracer techniques in general? While the technique discussed is applicable for assessing potential ecosystem cobenefits it is less applicable to CDR. Perhaps details on He/SF6 and gas exchange for example from the SOGasEx experiments would be useful (e.g. Ho et al., 2011).

We have opted to keep this section focused on the dual tracer technique for tracking alkalinity additions as we feel that Table 3 offers readers some insights into these other techniques.

Table 3. It might be worth mentioning the very high global warming potential of the 2 gases in this table as a potential limitation.

Added.

L320-335 There are many references that could be added here on the diel, seasonal and interannual variability/controls of carbonate chemistry and air sea gas exchange (e.g. Bates et al., 1998; Bates, 2002; Hagens and Middelburg, 2016; Landschützer et al., 2018; Sutton et al., 2019; Torres et al., 2021). Often these studies relate to the "time-of emergence" of anthropogenic trends but the similar signal-to-noise considerations are relevant for CDR detection.

Citations added.

L320-335 numerical simulations and machine learning based network design are potentially valuable tools to optimize observational networks to detect changes.

Added text:

L396: Numerical simulations and machine learning based network design are potentially valuable tools to optimize observational networks to detect experimental change.

**Minor points**

L52. Maybe "functionality" should be "efficacy".

"Efficacy" was added to the text.

L235. Should "categorizing" by characterizing?

Changed.

L272. I wouldn't refer to air-sea gas exchange as a community carbon flux (it would occur in abiotic waters).

"Community" was removed from the text.

**References**

Bates, N. R.: Seasonal variability of the effect of coral reefs on seawater $CO_2$ and air—sea $CO_2$ exchange, Limnology and Oceanography, 47, 43–52, https://doi.org/10.4319/lo.2002.47.1.0043, 2002.

Bates, N. R., Takahashi, T., Chipman, D. W., and Knap, A. H.: Variability of $pCO_2$ on diel to seasonal timescales in the Sargasso Sea near Bermuda, Journal of Geophysical Research: Oceans, 103, 15567–15585, https://doi.org/10.1029/98JC00247, 1998.

Egleston, E. S., Sabine, C. L., and Morel, F. M. M.: Revelle revisited: Buffer factors that quantify the response of ocean chemistry to changes in DIC and alkalinity, Global Biogeochem. Cycles, 24, GB1002, https://doi.org/10.1029/2008GB003407, 2010.

Hagens, M. and Middelburg, J. J.: Attributing seasonal pH variability in surface ocean waters to governing factors, Geophys. Res. Lett., 43, 2016GL071719, https://doi.org/10.1002/2016GL071719, 2016.

Hauck, J., Köhler, P., Wolf-Gladrow, D., and Völker, C.: Iron fertilisation and century-scale effects of open ocean dissolution of olivine in a simulated $CO_2$ removal experiment, Environ. Res. Lett., 11, 024007, https://doi.org/10.1088/1748-9326/11/2/024007, 2016.

He, J. and Tyka, M. D.: Limits and $CO_2$ equilibration of near-coast alkalinity enhancement, Biogeosciences, 20, 27–43, https://doi.org/10.5194/bg-20-27-2023, 2023.

Ho, D. T., Sabine, C. L., Hebert, D., Ullman, D. S., Wanninkhof, R., Hamme, R. C., Strutton, P. G., Hales, B., Edson, J. B., and Hargreaves, B. R.: Southern Ocean Gas Exchange Experiment:

Setting the stage, Journal of Geophysical Research: Oceans, 116, https://doi.org/10.1029/2010JC006852, 2011.

Landschützer, P., Gruber, N., Bakker, D. C. E., Stemmler, I., and Six, K. D.: Strengthening seasonal marine CO 2 variations due to increasing atmospheric CO 2, Nature Climate Change, 8, 146–150, https://doi.org/10.1038/s41558-017-0057-x, 2018.

Renforth, P. and Henderson, G.: Assessing ocean alkalinity for carbon sequestration, Reviews of Geophysics, 55, 636–674, https://doi.org/10.1002/2016RG000533, 2017.

Sutton, A. J., Feely, R. A., Maenner-Jones, S., Musielwicz, S., Osborne, J., Dietrich, C., Monacci, N., Cross, J., Bott, R., Kozyr, A., Andersson, A. J., Bates, N. R., Cai, W.-J., Cronin, M. F., De Carlo, E. H., Hales, B., Howden, S. D., Lee, C. M., Manzello, D. P., McPhaden, M. J., Meléndez, M., Mickett, J. B., Newton, J. A., Noakes, S. E., Noh, J. H., Olafsdottir, S. R., Salisbury, J. E., Send, U., Trull, T. W., Vandemark, D. C., and Weller, R. A.: Autonomous seawater $p$CO$_2$ and pH time series from 40 surface buoys and the emergence of anthropogenic trends, Earth System Science Data, 11, 421–439, https://doi.org/10.5194/essd-11-421-2019, 2019.

Torres, O., Kwiatkowski, L., Sutton, A. J., Dorey, N., and Orr, J. C.: Characterizing Mean and Extreme Diurnal Variability of Ocean CO2 System Variables Across Marine Environments, Geophysical Research Letters, 48, e2020GL090228, https://doi.org/10.1029/2020GL090228, 2021.

**Reviewer 2:**

Overall I found this chapter to be well-written and quite comprehensive. As someone at more advanced stages of planning such a field trial, I found some concepts to be somewhat obvious, but this is likely just my personal circumstance. For someone at the early stages of considering or even planning a field trial, I see this being a very valuable guide. I recognize the challenging in presenting such a guidebook at such an early stage, as there are little-to-no lessons/examples to build from. Overall, I have a series of minor comments but feel this manuscript 'does the trick' as a best-practices guide, at least at this early stage.

We thank the reviewer for their constructive comments and suggestions. Overall, we agree it is difficult to make concrete recommendations based on the few OAE field trials that are still ongoing. In our revisions we added a "Key Recommendations Section" which provides some more concrete recommendations based on the reviewers' comments.

General comments are as follows:

Little mention of emissions accounting and associated logistics are discussed here, but gaining an accurate picture of the LCA behind a proposed OAE deployment could be a critical learning objective. Not sure if this deserves mention in a sentence, or perhaps an entire section. Perhaps this is discussed more in another part of the Best Practices guide, but i think it should be mentioned more in this chapter.

We added more discussion on life cycle assessment and the roles it can play in determining the efficacy of the OAE projects and highlight the, pointing the reader towards other discussions and studies.

L68: "Life cycle assessments (LCA) may be a critical learning objective for some projects (e.g., Foteinis et al. 2023), especially those that are examining OAE at the scale of operational deployments."

L247: "Logistics will ultimately determine where operational OAE deployments take place and early field experiments will help to elucidate important issues including the impacts of life cycle emissions on CDR."

Given how early-stage we are as a community, one general note of caution is about using language like 'essential'. Generally the authors do a good job avoiding this, but one example in on line 334. Good baselining of a region is deemed 'esssential', but that this may take years to do. Perhaps this idea of 'good baselining' is intentionally vague, and ultimately I agree that baselining is important, but I feel that the assertion that years of baselining is required can create a signifiant barrier to progress. In my view, some level of baselining is done prior to a field experiment, but understanding seasonal and/or interannual changes in the region could occur during the progression at a proposed pilot site (i.e. after initial trial(s) have occurred). In contrast, line 219 states that it may or may be a 'priority' to measure trace metals or nutrient concentrations. I think labeling things as 'priorities' is a better way to go.

We have changed the language to reflect this comment and have adopted the verbiage of research 'priorities' throughout. Specific changes relevant to this comment are:

L381: "Gaining baseline knowledge on the physical, chemical, and biological components of the study site should be a priority. There is often considerable natural variability in marine systems, and especially in coastal systems, due to fluctuations in biological activity, hydrodynamics, seasonal and/or interannual influences, and others (Bates et al., 1998; Bates 2002; Hagens and Middelburg, 2016; Landschützer et al., 2018; Sutton et al., 2019; Kapsenberg and Cyronak, 2019; Torres et al., 2021). Fully characterizing this variability could take many years, which may create significant barriers to experimental progress in the field. Therefore, we recommend that any potential spatiotemporal variability be recognized and evaluated while planning field experiments."

It may be worth recognizing early in the document that resources will quickly become limiting in these early field trials if attempting to 'do it all'. The authors mentioned this when establishing main goals of the experiment, but explicitly stating that each of these 'potential overarching goals' listed are large and complex endeavors might be wise. The best example of this of course is the community engagement piece of a field trial, very easy to say, but very challenging to do properly, and way out of the wheelhouse of academic/industry researchers that often lead experimental development.

We agree that resources can become limiting and it is important to focus field trials on specific goals and outcomes. Collaboration with experts in aspects like community engagement that are outside the expertise of researchers will be an important part to the success of OAE and we tried to highlight this point in our revisions.

The article only has essentially 1 figure, albiet a very useful one showing conceptual dye tracer work. Given context of the material a lack of figures is understandable, but more of these idealized figures could have been useful. One example could be a visual showing a theroteical alkalinity plume and how one might most effectively sample that plume (using different sensors/platforms)?

We have opted to refer the reader to Figure 1 in Bushinsky et al. (2019) which shows the spatial and temporal capabilities of different sampling schemes for ocean carbonate chemistry.

L288: "Seawater carbonate chemistry measurements will be central to most sampling schemes. To cover the appropriate spatial and temporal scales, traditional bottle sampling will likely have to be combined with state of the art in situ sensors (Bushinsky et al., 2019; Briggs et al., 2020; Ho et al., this Guide). Bushinsky et al. (Figure 1; 2019) provides a comprehensive overview of the spatiotemporal capabilities of existing carbonate chemistry sensors and platforms."

I have a number of relatively minor suggestions/comments, but overall I commend the authors of a job well done. Suggestions, line by line, are given below:

Line 38: LCA can be a critical logistical contraint.

Added to list.

Line 59: may not be appropriate for a project to only focus on one of these goals. Seems like responsible (and publicly acceptable) research may want to prioritize having at least elements of 3+ of these larger goals.

We agree for larger projects, but small projects may be highly focused sacrificing responsibility/acceptance. We have removed the text 'one goal' but opted to keep 'highly focused'.

Line 64: 'require' tracking both the dissolution PLUS the fate of the dissolved alkalinity...I think it's feasible that a project focus on the dissolution and not try and track the fate. Again, words like prioritize feel more appropriate here.

Text has been updated to reflect this:

L78: "For example, projects adding ground alkaline minerals (e.g., olivine) to the ocean may have different goals and timelines than projects that add aqueous alkalinity (e.g., liquid NaOH) (see Eisaman et al., 2023, this Guide). Priorities for projects adding ground material could include tracking the dissolution of the alkaline material plus monitoring the fate of the dissolved alkalinity and its dissolution co-products (e.g. trace metals), while projects adding aqueous alkalinity will likely be concerned more with the fate of alkalinity."

Line 68: residence time of the receiving waters is an important consideration (perhaps covered by dilution and advection)

Added.

Section 2.1: important to note that these alkalinty types fall along a spectrum, and are not binary between rocks that dissolve slowly and electrochemical techniques. Some rocks/minerals dissolve very quickly and thus are probably closer to electrochemical than to olivine.  This is important because measuring in-situ dissolution will be very challenging when using a rapidly dissolving mineral in the water column. This is discussed more later in the document (e.g. line 165-166), and in the tables, but at this early point it reads like just 2 types and nothing in between.

We have opted to leave this more nuanced discussion for later in the document, as we believe there are distinct differences between adding material and electrochemical alkalinity additions.

Line 104: just a note that 'tracking the added alkalinity' and 'observing drawdown of atmospheric CO2' are very different things. I'd arge they are major challenges 2 and 3. Later in the paragraph its stated 'this is likely to be the main scientific concern'. I feel these need to be split out because directly measuring the CO2 uptake is probably not going to be main concern early on when the alkalinity addition rates are small.

This wording has been changed:

L128: "The second major challenge is common to both solid and aqueous approaches and involves tracking the added alkalinity, which becomes a particularly difficult problem in open-system field experiments where water is freely exchanged. Depending on the objectives of the field deployment, this is likely to be a main scientific concern. However, it is important to note that tracking the added alkalinity does not necessarily equate to observing CDR, or an increase in seawater $CO_2$ stored as bicarbonate or carbonate. Observing an increase in atmospheric $CO_2$ stored as seawater dissolved inorganic carbon comes with its own set of challenges that are discussed in depth by Ho et al. (2023, this Guide)."

Line 109: we aren't generating a 'DIC deficit' right that is then equilibrated, right?

See previous comment.

Lines 118-129: some of this felt a bit obvious, but may be a biased view

We have added some additional text to this section.

Line 140-142: this feels obvious but is important. Lets not waste too much time/resources on scenarios that aren't going to be applicable down the road. Two lines later the authors suggest using 'barriers' to avoid rapid loss of ground material...to me its not an open system anymore and that is a problematic for applying results down the road.

We have updated the text:

187: "Experiments could also artificially contain the material using barriers to avoid rapid loss of the ground material via currents, however, this could make the experiment less comparable to real world OAE deployments."

Line 155: authors could suggest that 'guardrails' are put in place that explicitly state what types of results would generate a 'pause' in the experiment. That seems to be well received as an idea.

This is a great comment and we have added text to reflect this:

L201: "Safety criteria should be put in place that can create a pause in the field experiment or prevent future experiments of the same type from taking place. These guardrails should be developed by the broader OAE community but may include obvious damage or health impacts to ecologically important organisms such as primary producers and keystone species, large and unexpected changes in biogeochemical cycles, and the general deterioration of environmental conditions. Risk-benefit analysis may be particularly useful in determining whether projects can or should move forward and may already be included in regulatory requirements through existing frameworks such as environmental impact assessments."

Table 1: I feel it should read "natural or manufactured magnesium-derived alkalinity sources", rather than just $Mg(OH)_2$. Just putting $Mg(OH)_2$ limtis you to 'reagent grade' $Mg(OH)_2$, which I doubt will be used too often at field-study scale.  There are many 'manufactured' $Mg(OH)_2$

products just like the lime-derived sources (these would have trace nutrients and metals just like the CaO). Also, you could put Mg(OH)2 and CaO etc. as their own rows, but stikcing to Mg-dervied and lime-derived might be easier.

Changed.

Line 195/Box1: i feel there are geological considerations too. Sediment type, underlying geology, in some cases must play a role? Also, water depth is quite important. No explicit mention of 'natural variability' in the system. No mention of 'prior data', how much exists to date is an important consideration.

We opted to keep these discussions in the main text.

Table 2 seems well written and quite comprehensive. In my experience, some communities want more thought beyond planktonic and benthic species...as challenging as that can be.

Line 261-263:  A little confused. Aren't small-scale experiments still directly transferable to to natural systems, and they help us learn prior to larger scale deployments. I suppose in some cases the early field trials may not be as transferable, but i'd advise against that as much as is reasonable.

We reworded to 'these types of studies' to address this comment.

Line 271: The dual tracer example/technique is very nicely presented and will be very helpful. I wonder if there are other specific examples like this that could be added/explored. I can't think of any just now but worth a thought.

Line 323: as mentioned at the outset. 'essential' multi-year baselining (even if not written exactly like that) could be a siginificant barrier. Could a suggestion be made that some baselining must occur, and can/should continue at the early stages of a site development (because the alkalinity additions will eventually fully flush, leaving the system ready for decent baselining again? A fairly repetitive statement like this is on line 334-335.

Updated based on previous comments.

Line 379-380 : I appreciate how the authors provide suggestions alongside actual ideas/examples. For example, the idea of a centralized data platform is backed up by two potential ideas for where such a platform could be housed.

Thank you for your helpful comments.

**Reviewer 3:**

This chapter provides a decent starting point for those considering undertaking an OAE field trial. I believe it nicely conveys the breadth of considerations and there is only one major piece of the puzzle missing from the discussion, namely logistics. I also believe it would benefit from some clearer definitions around terms like "field trial" and "full-scale". I detail these two concerns below followed by some minor comments. In general, this chapter should be published following the appropriate revisions.

We appreciate the reviewer's positive comments and have addressed them as outlined below.

 Comment 1: Logistics – This chapter focused primarily on scientific considerations with respect to field trials but appropriately acknowledged permitting and stakeholder engagement as well. However, there was no discussion of logistics and the significant role that plays in shaping field trials. For example, I believe proximity to a marine institute (for land-based approaches) or access to a research cruise (for open ocean) is often essential to execute the type of science program advocated for here. In both cases, the location of these kinds of scientific resources plays a defacto but significant role in dictating where one does a field trial.

We added text to reflect this:

L244: "These considerations will grow with the scale of field experiments and will likely be first-order determinants of where field experiments take place. For example, proximity to a marine institute (for land-based approaches) or access to a research cruise (for open ocean approaches) may be desirable."

Logistics can manifest in other ways too that may be obvious but weren't stated here: If your OAE approach is tied to wastewater discharge or desalinization, then you're constrained to areas with these plants. If your OAE approach is fundamentally about putting sand on coastlines then you need to work on stretches of coastline that have roads and truck access, or have deep enough water that they are accessible by dredge so that you can actually place the material. If your reactor process consumes lots of energy, you may want to think about the local grid before you start building. The list goes on.

We agree and have amended the text to reflect this:

L247: "Logistics will ultimately determine where operational OAE deployments take place and early field experiments will help to elucidate important issues including the impacts of life cycle emissions on CDR."

Comment 2: Definitions – I believe this chapter would benefit from a more thorough discussion of what is actually meant by "field trial" versus "full-scale." The authors loosely define field experiment as "the addition of alkalinity to a natural system in ways that simulate planned OAE deployments" but I find this vague. What is the difference between something that "simulates" a deployment and an actual deployment? Is a field trial defined by the size of the intervention –

total added alkalinity, length of time alkalinity is added, both? What is the line? Is "small" or "big" different for different kinds of OAE interventions?

Perhaps "field trial" is defined by the extent of the rigorous, in-depth science program? Maybe it is defined by the project's owners or goals (e.g. academic or NGO research, industry R&D, carbon credit sales) regardless of the breadth of the science program?

Alternatively, is "field trial" defined by the permit as in many jurisdictions there are specific "research permits" that can be sought?

A combination of above? Who gets to decide and why? What are the implications? What is precedent from other areas of CDR, CCS?

Anyway, I think this is important because depending on how we define "field trial" that of course can change the guidance on how to conduct one. As an example, if field trials inherently have to be small-scale, the authors should provide some thinking on how to define "small" for any given intervention and it fundamentally changes the discussion about potential environmental impacts associated with field trials. Statements such as "field experiments evaluating CDR approaches carry the risk of unintended consequences and impacts over vast spatial scales, so appropriate scaling (e.g., starting small) is necessary" (pg 20) are no longer appropriate. Rather, the small-scale nature of field trials is then key to their value in that they can evaluate the potential for impact without meaningful risk of causing significant impact over larger spatial or temporal scales themselves. I think that is an important point about starting small that I would love to see emphasized in the text.

As a side note, do the authors distinguish between "field trials" and "field experiments"? Both are used interchangeably in the text but the common definition of a field trial is a test of a product or device in the environment while the common definition of a field experiment is simply an experiment conducted in the natural environment (but not necessarily of a product; the product here being the OAE intervention). If field trials and field experiments are different then how does that change the guidance in this chapter for one versus the other? In the introduction the authors define 'field experiment' as "the addition of alkalinity to a natural system in ways that simulate planned OAE deployments" but elsewhere in the text make statements such as "to make results more broadly applicable, field experiments should attempt to mimic real world alkalinity application scenarios" (pg 6) and "field experiments will presumably mimic plans for

real world OAE deployments." These statements seem redundant with the authors given definition.

We have updated the introduction to discuss these definitions explicitly. We have also amened the text to reflect these new distinctions throughout the manuscript.

L31: "This chapter addresses considerations for conducting open-system field experiments related to ocean alkalinity enhancement (OAE). We define 'field experiment' or 'field studies' broadly as the addition or manipulation of alkalinity in a natural system that is relevant to OAE, independent of the spatial and temporal scale. We intentionally exclude spatial and temporal scales from our definition to encompass the wide spectrum of OAE methods and approaches. In fact, field experiments are likely to span spatial scales of m2 to 100s of km2 and last from days to years. Field experiments and studies differ from both 'field trials' and 'field deployments' in their motivation, as both trials and deployments denote the practical application and usage of a specific product, device, or technology. The scientific focus during field trials is likely to be on the efficacy of Carbon Dioxide Removal (CDR) and fine-tuning operational deployment, while field experiments will encompass a broader range of scientific goals and objectives. The nature, logistics, and objectives of field experiments are likely to make them smaller in scale than operational deployments. This will be advantageous, as field experiments that emulate planned OAE trials and deployments will help create the scientific framework needed to scale operational OAE safely and responsibly."

Minor Comments:

L64: Projects adding alkaline materials to the ocean are not "required" to track the dissolution of those materials, it may be sufficient to track one or the other.

Changed.

L70: This section starts off as "addressing public concern". The framing should be broader here. Not all members of the public will be concerned; some will be supportive, some will be neutral but they must all be given an opportunity to be informed and have their questions answered. Same comment L358.

These sections have been updated:

L90: "Addressing the appropriate regulatory requirements is essential before any field experiment can move forward. Permitting requirements will be influenced by the study location, type of alkalinity perturbation, spatial scale, and duration. The use of existing infrastructure (e.g., wastewater discharge sites) and environmental projects (e.g., beach renourishment) may offer ways to facilitate alkalinity perturbations under existing regulatory frameworks. Community engagement and outreach are other areas that will be important to address, especially when the alkalinity perturbation is large and uncontained. Ideally, local communities should be engaged at the earliest possible stage since social license to operate is critical for the success of CDR

projects (Nawaz et al., 2022). For a more detailed discussion of legal and social issues see Steenkamp et al. (2023, this Guide) and Satterfield et al. (2023, this Guide)."

L71: The field site (e.g. state and/or nation-level boundaries, for example) is a factor in determining permitting requirements but so is the source of alkalinity and method of introduction.

Text was changed to address this.

L91: "Permitting requirements will be influenced by the study location, type of alkalinity perturbation, spatial scale, and duration."

L109: Not a DIC deficit but a CO2 deficit & and also not necessarily a deficit, OAE also works when outgassing is prevented (L114).

This was addressed in the text.

L120: Also important for minimizing environmental impact and in many cases, meeting regulatory requirements.

Added text:

L153: "Other experimental considerations related to the type of alkalinity perturbation include the duration and location of alkalinity addition, which will important for environmental and regulatory considerations."

L153: I would argue that best practice should include testing to insure there are no "other contaminants."

 We agree.

L150-179: I think this page would benefit from a discussion on how potential environmental impact changes with the scale of the field trial and also how to evaluate risk. For example, permits in the US require documents like Environmental Assessments which consider both the potential impact but also the likelihood that the impact is sustained, efforts to minimize the impact, efforts to monitor the impact, and whether the risk outweight the benefit of the project. Also, many potential impacts of OAE additions that are referenced in this chapter (e.g. turbidity for coastal enhanced weathering and extreme localized changes in carbonate chemistry for aqueous additions) are evaluated in standard permitting pathways (e.g. beach renourishment and wastewater discharge, respectively), so regulatory compliance will help insure these potential impacts are addressed. In summary, impact is nuanced and a risk-benefit framework is generally used. I would advocate for this chapter to provide guidance on risk-benefit analysis. Foteinis et al., 2023 also has some nice thinking on this.

 We have added the following text to address this:

L201: "Safety criteria should be put in place that can create a pause in the field experiment or prevent future experiments of the same type from taking place. These guardrails should be developed by the broader OAE community but may include obvious damage or health impacts to ecologically important organisms such as primary producers and keystone species, large and unexpected changes in biogeochemical cycles, and the general deterioration of environmental conditions. Risk-benefit analysis may be particularly useful in determining whether projects can or should move forward and may already be included in regulatory requirements through existing frameworks such as environmental impact assessments."

Table 1: Naturally-occurring brucite (as opposed to synthetic MgOH2) contains other elements as well but the two are not distinguished here.

Updated.

Box 1, Biology: I would add the consideration of whether there are culturally or commercially important species present. Not that the presence of commercially important species would necessarily be an issue, but one might want to be sure to have a focus on that in the monitoring program.

Added.

L333: Im not sure how practical it is to suggest that baselining occur "over long periods of time". This is probably another factor that should be incorporated into site selection, i.e. select locations with a wealth of pre-existing scientific data either in the peer-reviewed literature and/or proximity to publically available (often government run) stations/buoys such as the USGS or NOAA monitoring networks in the US.

Added text:

L259: "Due to the large investments in cost and time required to collect baseline data, locations with a wealth of pre-existing scientific data may be considered. This baseline data could be available in the peer-reviewed literature and/or from publicly available coastal and open ocean time-series (e.g., Sutton et al., 2019)."

L371: I would include a call to publish data in open source, peer-reviewed journals whenever possible.

Added:

L436: "When publishing in peer-reviewed literature, uploading data to publicly available data repositories and publishing in open access journals following best practices should be prioritized (Liang et al., this Guide)."

**Reviewer 4:** This is a well written overview of the techniques, pros and cons of open field experiments. It will provide a valuable resource to those planning open field experiments. There are a few minor issues that should be addressed.

Ln 62-63: Is it worth also highlighting the need to monitor the sinking speeds of the (ground) particulate material?

This is addressed later in the manuscript.

Ln 158-159: Another important consideration to highlight is the potential for bio-accumulation in higher trophic level organisms (especially those of commercial importance).

Sentence added:

L200: "The bioaccumulation of toxic metals in higher trophic level organisms, especially those of commercial importance, is a particularly important concern."

Box 1: Chemistry, 'or micronutrients' – some of these micronutrients are also potentially toxic at high concentrations.

Wording changed to reflect this.

Table 2: typo in carbonate chemistry conditions row – how much rather than high much.

Changed.

Table 2: macronutrient and micronutrient rows – Beyond the potential pathway for assessment of the basic methods for determining concentrations of these elements, should the table highlight experimental methodology to allow the 'assessment of whether the designated system is prone to' macronutrient or micronutrient fertilization via OAE.

Added "Experimental assessment of limiting elements" to the text.

Ln 237: 'ideally' – this seems an understatement; surely in any OAE experimental work that needs to understand the efficacy and impact of OAE, two carbonate chemistry parameters should be measured as standard.

Removed 'ideally'.

Ln 268: 'additional alkalinity' – do you mean excess alkalinity?

Changed to 'added'

Fig 2. Differences between panels (a) and (b) are too subtle and it is difficult to distinguish. What about putting the panels together and emphasizing the differences in some way?

We opted to keep this figure the same because it is based on a previous publication.